# Control of Translation at the Initiation Phase During Glucose Starvation in Yeast

**DOI:** 10.3390/ijms20164043

**Published:** 2019-08-19

**Authors:** Yoshika Janapala, Thomas Preiss, Nikolay E. Shirokikh

**Affiliations:** 1EMBL-Australia Collaborating Group, Department of Genome Sciences, The John Curtin School of Medical Research, The Australian National University, Canberra, ACT 2601, Australia; 2Victor Chang Cardiac Research Institute, Darlinghurst, NSW 2010, Australia

**Keywords:** nutrient stress, glucose starvation, stress response, eukaryotic translation, eukaryotic protein synthesis control, rapid response to stress, translation mechanisms, translation initiation, mRNA, mRNP, ribosome, eIF, 5’UTR, UTR, stress granules

## Abstract

Glucose is one of the most important sources of carbon across all life. Glucose starvation is a key stress relevant to all eukaryotic cells. Glucose starvation responses have important implications in diseases, such as diabetes and cancer. In yeast, glucose starvation causes rapid and dramatic effects on the synthesis of proteins (mRNA translation). Response to glucose deficiency targets the initiation phase of translation by different mechanisms and with diverse dynamics. Concomitantly, translationally repressed mRNAs and components of the protein synthesis machinery may enter a variety of cytoplasmic foci, which also form with variable kinetics and may store or degrade mRNA. Much progress has been made in understanding these processes in the last decade, including with the use of high-throughput/omics methods of RNA and RNA:protein detection. This review dissects the current knowledge of yeast reactions to glucose starvation systematized by the stage of translation initiation, with the focus on rapid responses. We provide parallels to mechanisms found in higher eukaryotes, such as metazoans, for the most critical responses, and point out major remaining gaps in knowledge and possible future directions of research on translational responses to glucose starvation.

## 1. Introduction

In order to survive and/or proliferate under changing and stressful conditions, different organisms have evolved elaborate mechanisms for gene control [1,2,3]. To rapidly optimize their gene expression profiles, eukaryotes complement slower nucleus-based transcription-level reactions with cytoplasmic control of messenger(m)RNAs translation into proteins and dynamically alter the composition of the proteome [4]. This allows for a rapid response to stresses and can often initiate a longer-term transcriptional reprogramming, to change gene expression profiles more permanently [5,6]. Another important reason to rapidly alter the rates of protein production is that translation is one of the most nutrient and energy-costly processes for most cells [7,8]. Thus, failure to quickly adjust protein synthesis levels may lead to an inability to produce proteins critical for ongoing homeostasis [9,10].

Yeast cells are known to respond at the level of translation to a multitude of environmental stresses, such as heat shock [11,12], nutrient deprivation [13,14,15], general amino acid balance [16,17,18,19], phosphate [20], sulfur [21,22], nitrogen [23,24] content and other environmental conditions [25,26], such as growth at low/high temperatures [27], in different oxygen levels [28,29,30] or in the presence of heavy metals, during desiccation [31,32,33], etc. Many of these conditions are studied in the context of industrial applications of yeast in bread baking and alcohol fermentation [34,35,36,37,38].

Of all nutrients, d-glucose (‘glucose’) is by far the most important, and the favorite carbon and energy source in use by *Saccharomyces cerevisiae* (hereinafter ‘yeast’ unless indicated otherwise) [39]. Glucose starvation causes a rapid, but also rapidly reversible, inhibition of translation in yeast by mechanisms that are surprisingly not as well elucidated as some other nutrient responses [40,41]. Several groups have tested the time (duration of starvation) and the reduction in amount (concentration) of glucose needed to induce this rapid effect in experiments at various stages of yeast culture growth, e.g., References [40,42,43,44] (see Table 1 and Figure 1 for summary and Table 2, Table 3, Table 4 and Table 5 and Figure 2 and Figure 3 for more detail). They used approaches such as polyribosome (polysome) complex separation by centrifugation through a sucrose gradient (polysome profiling), high-throughput sequencing of ribosome-protected mRNA fragments (ribosome profiling) and immunoprecipitation of RNA-protein complexes. When glucose levels decreased below 0.6% (*w*/*v*), an acute inhibition of translation was observed as indicated by the decrease in cytosolic polysome content and a concomitant increase in single ribosomes (monosomes), an effect which can be reversed within minutes (min) by re-supplementation of glucose [40,42]. Another prominent reaction to glucose depletion is an (often selective) inclusion of mRNA into a variety of cytoplasmic foci or granules, often through ‘phase separation’ effects [45,46,47,48], whereby the mRNA becomes generally translationally repressed and may also be degraded [49,50,51,52,53,54] (Table 5 and Figure 4).

The importance of glucose and the physiological or pathophysiological responses to its levels, however, extend beyond yeast to other eukaryotes, with a considerable degree of conservation [65,66]. Translational responses to glucose starvation are an integral part of mammalian metabolic control and become critical under physical stress (exercise) and in diseases, such as diabetes or cancer [67,68]. Using the wealth of data for glucose starvation responses in yeast, we structure this review to first discuss the translational mechanisms of glucose starvation response as they are elucidated for yeast and then briefly comment on mammalian- or higher-eukaryote-specific pathways in a comparative fashion. We then summarize glucose starvation responses leading to mRNA inclusion into cytoplasmic granules.

## 2. Overview of Eukaryotic Translation Initiation

Translation (e.g., reviewed in References [8,69,70,71,72,73,74,75]) is a complex process conserved across all life, but with several eukaryote-specific features [76,77]. Highly conserved aspects include ribosomes, transfer(t)RNAs, the genetic code and the division of translation into distinct phases, including initiation, elongation and termination of polypeptide synthesis, as well as recycling of the ribosome for subsequent translation rounds. With very few exceptions, a new round of translation on mRNA is initiated by the ribosomal small subunit (SSU), which together with specialized initiation factors (IFs) locates the start codon of an open reading frame (ORF) within an mRNA [8,71,78,79,80]. Initiation completes by joining the ribosomal large subunit (LSU) with the SSU to form an elongation-capable complete ribosome [71,80,81,82,83]. Peptide bonds are generally formed as specified by the codon sequence during elongation, which is perhaps the most conserved phase of translation. Termination is also well-conserved and involves the release of the synthesized polypeptide from the ribosome when the stop codon is reached. Finally, ribosomes are (sequentially) disassembled into subunits and are charged once again with IFs in the process of recycling [69,84,85,86,87].

Eukaryote-specific features can be found throughout all phases of translation, but the most profoundly modified phase is initiation (Figure 1). This is reflected at the level of mRNA in the eukaryote-specific determinants, the 5’ cap structure and 3’ poly(A) sequence, and at the level of auxiliary factors in that they include the poly(A)-binding protein (Pab1p in yeast) and a multitude of eukaryotic IFs (eIFs) that either lack prokaryotic homologs or they are not universally involved in prokaryotic translation initiation [36,75,88,89]. The most distinctive mechanistic feature of eukaryotic initiation is the ‘scanning’ process by which the start codon is located on mRNA [90,91]. In prokaryotes, initiator formyl-methionyl-tRNA (fMet-tRNA^fMet^) associates with the SSU [92] and, with few exceptions [93,94], the start codon is identified through direct base pairing of the Shine–Dalgarno sequence, located upstream of the AUG in the mRNA, to a complementary sequence near the 3’ end of the SSU ribosomal RNA [95,96,97,98,99,100,101]. By contrast, in eukaryotes there is no dedicated anti-Shine–Dalgarno-like sequence in ribosomal RNA, although functionally similar mRNA-to-ribosomal RNA base-pairing has been proposed for a few mRNAs [102,103]. Instead, SSUs are directed to the 5’ end of the mRNA by virtue of an eIF-mediated affinity to the cap, and then move along mRNA in 3’ direction [71,104,105,106,107]. The 5’UTR sequence is continually probed for the presence of a start codon during this ‘scanning’ process and, as a result, the most 5’-proximal start site in a strong context is usually selected [75,86,90,91].

The mRNA cap structure and 3’ poly(A) have strong stimulatory effects on initiation, mediated by their respective binding proteins eIF4E and Pabp1. This is thought to involve mRNA ‘closed-loop complex’ (CLC) structures formed through cap:eIF4E:eIF4G:Pab1p:3’ poly(A) bridging interactions (Figure 1) [108,109,110,111,112]. The mRNA CLC enhances SSU attachment to the cap, but also has effects beyond that, such as in re-initiation, ribosome recycling [113] and mRNA stability (reviewed in Reference [114]). The binding of Pab1p to poly(A) increases the affinity of Pab1p for eIF4G and the affinity of eIF4E for the cap, which in turn increases translation initiation efficiency [115]. CLC function in translation is incompletely understood. In live yeast, different mRNAs exhibit different levels of CLC prevalence. Thus, it is thought to be a dynamic, part-time interaction in vivo, rather than a stable and continuous structural arrangement [108,116]. The initiation can also occur in a cap-independent manner [117,118,119,120,121]. Usually, cap independence is conferred through internal ribosome entry sites (IRESs) found in certain viral genomes [122,123,124,125], but also some cellular mRNAs. Translation of IRES-containing mRNAs can be favored during environmental changes (stress, cell division, apoptosis) that reduce global cap-dependent translation (reviewed in References [126,127,128,129,130]). There are multiple examples of alternative translation initiation routes, including eIF4E-independent translation [131], eIF4F/3-independent initiation [78,132,133], initiation on leaderless mRNA [134,135], etc., (e.g., for review, see Reference [75]). However, for an ‘average’ eukaryotic 5’UTR initiation is notoriously cap-dependent [136,137], and even a leaderless mRNA may exhibit a strong dependence on cap [138].

An SSU complex capable to commence scanning is thought to include at least eIFs 1, 1A, 2:GTP, 3, 4E, 4G and contain aminoacylated initiator tRNA (Met-tRNA_i_^Met^) (Figure 1) [8,72,75,86,103,139]. eIF2, Met-tRNA_i_^Met^ and GTP form a highly stable ‘ternary complex’ (TC), the levels of which substantially affect initiation efficiency [140,141,142,143]. eIF3 promotes TC binding to SSUs [144,145,146], in part through forming a ‘multifactor complex’ (MFC) with eIF1, eIF5 and the TC [104,147,148]. However, the strict TC requirement for scanning-like motion can be loosened in distinct cases, such as during re-initiation after translation of regulatory upstream(u)ORFs [91,149,150,151,152,153,154,155,156,157,158]. Typically, though, the eIF4F complex (comprising eIF4E cap binding protein, the RNA helicase eIF4A, and eIF4G) attaches to the cap and in the presence of (an excess of) additional ‘scanning’ factors, such as eIF4A/4B (and 4H in mammals) [159,160,161,162,163,164] and optionally yeast Ded1p (Ddx3 homologue in mammals) [164,165,166], the SSUs are considered to move along the 5’UTRs unidirectionally nucleotide-by-nucleotide driven by ATP/NTP hydrolysis and carrying their entourage of eIFs along with them [71,75,79,167,168,169,170]. The ATP/NTP-dependent RNA-helicases eIF4A, and possibly Ded1/Ddx3/DHX29 [63,171,172,173,174,175], are considered to be responsible for the directionality of the motion. Likewise, these proteins help scanning SSUs to overcome structural barriers along the 5’UTR [63,78,172,173]. Therefore, structural impediments and the availability of the eIFs involved in scanning are thought to be the main determinants of scanning efficiency [167,176,177,178].

As the 5’UTR sequence is continuously inspected by the anticodon and point-to-point interaction capacities of eIFs and SSUs, strong binding of a start codon can trigger a scanning stall. This leads to a series of rearrangements to form a more stable SSU:mRNA complex where Met-tRNA_i_^Met^ is situated in the P-site of the SSU and its anticodon is base-paired with the start codon (often referred to as the ‘48 S’ initiation complex) [146,179] (e.g., see Reference [75] for review). Key determinants of start codon recognition efficiency are a strong start site (a combination of start codon triplet and its nucleotide context), a slow-down of scanning, such as by structure downstream of the start site [180], and the availability of the ternary complex [80,179,181,182]. Additionally, eIF1 and similar factors decrease recognition of weak start sites (which, competitively, in an mRNA-rich system may increase recognition of the strong start sites). In some cases, factors alternative to eIF2 and the canonical ternary complex are utilized [183,184,185,186]. Early on, SSU complexes at start codons induce eIF2-bound GTP hydrolysis (activated by eIF5) and reconfigure, such as that eIF5B:GTP replaces eIF2 in the complex. The eIF5B:GTP:Met-tRNA_i_^Met^-containing SSU complexes then are joined by the LSU, which triggers eIF5B-bound GTP hydrolysis and eIF5B release (this late step resembles the entire initiation process in bacteria) [187,188] to form elongation-capable ribosomes [189,190,191,192].

Hence, eukaryotes evolved a substantially more complex initiation pathway where several additional factors serve as effective targets for eukaryote-specific control and often provide signal integration from different inputs to alter translation globally [193,194]. Mechanistically, it also adds at least two new control points, during the attachment to the cap structure and during scanning, and involves input from additional and variable sequences of the 5’ and 3’UTRs [195,196]. Although stress affects all phases of translation [197,198,199,200], many stress-induced responses target the initiation phase [3,70,196,197]. The evolutionary importance of translational control during the initiation phase is in that it can provide means for a selective response without the need of changing protein-coding sequences and requiring only changes in the UTRs [198], a subject of rapid evolution which is a feature not universally available to the other phases of translation. The glucose starvation response is no different and, perhaps unsurprisingly, its effects on translation initiation can be naturally dissected into the influences over the major stages of initiation, such as cap attachment, scanning and start codon recognition (Figure 1; main text Section 3, Section 4 and Section 5), along with more global effects on mRNA localization and accessibility (Figure 1; main text Section 6), an approach we adopt in this review to systematize the material.

Despite its mechanistic simplicity, prokaryotic translation initiation is still frequently and effectively used for gene expression adjustments [195,201]. Detailing prokaryotic mechanisms is beyond the scope of this review; however, the most common cases are riboswitch-type arrangements where SSU binding to the start site is blocked by either RNA or protein:RNA structures [202,203,204]. When these riboswitch structures are made dependent on smaller metabolic ligands, they can act in rapid nutrient responses (e.g., reviewed in References [195,201]). It is further worth noting that mitochondria (and chloroplasts) feature prokaryote-type translation initiation mechanisms and nevertheless form an integral part of coordinated eukaryotic cell responses [205,206]. The ribosome profiling approach [207] was recently modified for use with mitochondrial (mito)ribosomes to show that control of mitochondrial translation participates in the whole-cell glucose starvation response in yeast [208]. A shift from fermentable (2%) glucose to non-fermentable (3%) glycerol-based media led to a rapid and selective redistribution of mitoribosomes. Translational suppression of ATP synthase complex mRNAs and concomitant induction of cytochrome c oxidase complex mRNAs was observed 15 min after the media shift, followed by a return to near-normal levels by 1 h into the stress [208]. Most intriguingly, using cycloheximide to inhibit cytoplasmic, but not mitochondrial translation [209], it was determined that the cytoplasmic translational response leads the mitochondrial translational response, whereas there is no such interconnection at the level of transcriptional reprogramming [208]. These results demonstrate cross-compartment regulatory links of translational control that deserve further exploration, including how these links are enacted in multicellular organisms.

## 3. Regulation by Targeting Ribosomal Attachment to mRNA

eIF4F activity, or availability, and mRNA CLC prevalence, due to its synergism with cap-dependent translation, are the main pre-requisites of efficient ribosomal attachment to mRNA. In general, cap-dependent translation can be inhibited or outcompeted by means of (1) eIF4E regulatory proteins (4E-BPs); (2) phosphorylation of eIF4G by kinases; and (3) IRES-dependent translation. In the rapid yeast response to glucose starvation, eIF4G-based regulation has been proven to feature strongly, whereas rather surprisingly, 4E-BPs have not been shown to be critically important [74]. However, 4E-BP-based control may participate in the later stages of glucose response in yeast and in mammalian cells, it is a part of a universal translational nutrient response.

The key players regulating the assembly of the eIF4F complex in mammals are the eIF4E-binding proteins (4E-BPs), a family of repressor polypeptides sharing structural similarity with the eIF4E-binding region of eIF4G, around a YXXXXLϕ motif where ϕ is Leu, Met or Phe and X is any amino acid. By competing with eIF4G in binding to eIF4E, 4E-BPs inhibit eIF4F and CLC assembly and repress translation [143,210,211]. Binding of the 4E-BPs to eIF4E is regulated by eIF4E phosphorylation. Hypophosphorylated 4E-BP isoforms interact strongly with eIF4E, whereas hyperphosphorylated isoforms do not [212,213]. The mechanistic target of rapamycin (mTOR) is an evolutionarily conserved phosphatidylinositol-3 kinases (PI3K)-related protein kinase that forms two main complexes with different interacting partners and functions, TORC1 and TORC2. TORC1/2 regulate translation, cell growth and proliferation in response to nutrients, growth factors, ATP, oxygen levels, and other stressors [214,215,216,217,218,219]. Of the three mammalian 4E-BPs, 4E-BP1 is best-characterized and known to be phosphorylated by mTOR at Thr37 and Thr46 sites. The phosphorylation serves as a priming event for further phosphorylation of Ser/Thr/Pro sites, which severely diminishes 4E-BP1 ability to bind eIF4E [211,213,219]. Under optimal growth conditions, mTOR constitutively phosphorylates 4E-BP1 to prevent it from binding to eIF4E and inhibiting translation. In mammalian cells subjected to metabolic stress (including glucose starvation), inactivation of mTOR results in the accumulation of hypophosphorylated 4E-BP1, which binds eIF4E and inhibits translation [196,220] (Figure 2a).

Yeast encodes two known 4E-BPs, cap-associated factor 20 (Caf20p) and eIF4E-associated protein 1 (Eap1p). Both have no sequence homology with each other or with the metazoan 4E-BPs except for the highly conserved motif that contains YX4LΦ consensus sequence [221,222,223]. Caf20p and Eap1p regulate distinct but overlapping sets of mRNAs [224]. Both yeast 4E-BPs interact with eIF4E to inhibit translation of capped reporter mRNAs [221,222]. To test for the possible involvement of the yeast 4E-BPs in glucose starvation response, yeast cells (BY4741-derived) with either single or double deletions in the *Caf20* and *Eap1* genes (yMK1751, caf20Δ; yMK1752, eap1Δ; yMK1752, caf20Δ eap1Δ) were used (Table 2). Cells were starved for glucose in SC media for 10 min, and cell lysates assessed for polysome disassembly measured by UV absorbance traces after separation by sucrose gradient ultracentrifugation. Both the single and double mutant Caf20Δ/Eap1Δ strains exhibited the same dramatic alteration in polysome profile that was observed in the wild-type strain (yML1750) [42], presumably due to the ‘polysome run-off’ whereby translation elongation continues actively in the background of severely impaired initiation, resulting in rapid ribosomal clearance from mRNA and the disappearance of heavy polysomes [40]. Thus, 4E-BPs do not have substantial roles in rapid glucose starvation response in yeast, although longer-term effects controlled through these factors might still be possible [42].

In mammals, 4E-BPs were shown to regulate translation globally or in a more mRNA-specific manner. As revealed by ribosome profiling in mouse cells, inhibition of mTORC1 strongly impairs the translation of mRNAs containing 5’ Tract of Oligo Pyrimidine (TOP) sequences [216,225] (Figure 2c). Many components of the translational machinery, such as ribosomes and initiation factors, are thought to be TOP-containing mRNAs regulated by mTORC1 signaling. mRNA-specific inhibition of translation is demonstrated by other 4E-BPs with more direct effects, such as the classical examples of Cup (a 4E-BP):Bruno interactions in fruit flies to repress translation of oskar mRNA [226] and with Smaug to repress nanos mRNA translation [227].

However, the role of 4E-BPs in mammals is not very well studied specifically during glucose starvation, although 4E-BPs have extensive pathophysiological relevance in diseases such as diabetes [228]. For example, it is known that insulin can stimulate 4E-BP interactions with TORC1, and thus, possibly enhance cap-dependent translation, but glucose starvation effects of these mechanisms remain under-investigated [229]. Insulin activation of mTORC1 is accompanied by enhanced binding of substrates (4E-BP, S6 kinase (SK6) and Proline-Rich AKT1 Substrate 1 (PRAS40)) [229]. The mechanism of insulin activation of mTORC1 was studied in 293E and HeLa cells [229]. The cells were initially serum-starved in high glucose medium for 17 h (with an exchange for fresh media at 1 h to the end of the incubation), and then subjected to the fresh media with or without 1 μM insulin, for 30 min. The ability of mTORC1 substrates 4E-BP and PRAS40 to bind to Regulatory-Associated Protein of mTOR (raptor; part of the mTORC1) is strongly inhibited by their phosphorylation. Therefore, to render its association with raptor sensitive only to perturbations of mTORC1 or regulators of substrate binding to mTORC1, a mutant of 4E-BP fused to the carboxyl terminus of Glutathione S-Transferase (GST) wherein all five of the mTORC1-phosphorylated sites (Thr37, Thr46, Ser65, Thr70, and Ser83) were substituted with Ala (GST-4E-BP[5A]) was constructed. In 293E cells, depletion of the endogenous PRAS40 results in an increase in the amount of endogenous mTORC1 recovered with transiently synthesized GST-4E-BP[5A] from the cells that were serum-deprived and not stimulated with insulin, to ~70% of the level observed with the high insulin treatment where PRAS40 levels were not manipulated [229]. Because knockdown of PRAS40 in the presence of high insulin yielded no greater recovery of raptor with GST-4E-BP[5A] when compared to high insulin alone, PRAS40 depletion might increase substrate access to mTORC1, but not the mTOR signaling. In HeLa cells, high insulin and PRAS40 knockdown also enhanced the binding of 4E-BP[5A] to raptor, and high insulin alone stimulated S6K1 and 4E-BP phosphorylation. Thus, PRAS40 occupancy of raptor was concluded to be a major determinant of the ability of GST-4E-BP[5A] to bind to mTORC1, at least in response to insulin [229].

A negative feedback loop was described in metazoans for keeping basal mTORC1 activity low and tightly regulated and allowing growth when nutrients are limiting [230]. Phosphatidylinositol-5-phosphate 4-kinases (PIP4ks) specifically use phosphatidylinositol 5-phosphate (PI-5-P) to synthesize phosphatidylinositol 4,5-bisphosphate (PI-4,5-P2). The PIP4k family is well-conserved from worms to mammals, but absent in yeast [231]. While only one gene encodes PIP4k in *Drosophila* and *Caenorhabditis*, three distinct isoforms (α, β and γ) have been identified in mammals. The γ isoform is phosphorylated in response to growth factors [232] and was shown to be a substrate for mTORC1 involved in the regulation of cell size. Using HeLa and B cells from pancreas with insulinoma (BTC6 cells) grown in DMEM starved of glucose and glutamine for 3 to 5 h, it was shown that PIP4kγ knockdown impaired mTORC1 signaling. Further, in starved cells, it was shown that PIP4kγ exists predominantly in the unphosphorylated state and participates in the activation of mTORC1. When ATP and amino acids are abundant, and mTORC1 is fully active, phosphorylated PIP4kγ, which cannot activate mTORC1, becomes the predominant form. Therefore, the ratio between the phosphorylated and unphosphorylated forms of PIP4kγ depends on the activity of mTORC1 [230].

*Drosophila* larvae has been a versatile model for examining how nutrients control metabolism and growth during animal development. Larval growth in *Drosophila* is dependent on endocrine insulin signaling, as well as TORC1 signaling pathway [233,234,235]. In nutrient-rich medium, both pathways are activated, leading to cell, tissue and body growth, whereas nutrient stress leads to inhibition of insulin or the TORC1 signaling resulting in growth arrest. Polysome profiling experiments were performed to examine the influence of nutrient availability in *Drosophila* larvae on mRNA translation [236]. In the mid-point of larval growth at 72 h after egg laying, larvae were starved from fly food (agar, cornmeal, yeast, sucrose, glucose, propionic acid) by transfer to a 20% *w*/*v* sucrose solution in phosphate-buffered saline. Starvation of 30 min led to a reduction in the proportion of ribosomes engaged in translation and a concomitant increase in free SSUs, LSUs and mono-ribosomes. Starvation for 2 h continued this trend with lowest translation levels attained at 18 h, which did not decrease with further extensions of starvation (up to 4 days). Interestingly, while the levels of 4E-BP mRNAs remained high during all timepoints of starvation, ribosomal RNA abundance begun to decrease (compared to the non-starved control) only after 24 h upon starvation. Therefore, upon starvation, translation initiation may be inhibited as early as 30 min, but ribosome numbers decrease only after longer periods of the stress [236]. The *tor* null mutant larvae demonstrated reduced levels of polysomes generally comparable to the effects of starvation. However, TORC1 induction via overexpression of a gene encoding Ras homolog enriched in brain (Rheb) did not result in any recovery of the translation rates in the starved animals, suggesting that the response is more complex and might rely on more components and not just TORC1/4E-BPs, prompting further investigation [236].

Phosphorylation of eIF4G may affect cap attachment by the reduced formation of eIF4F complex, and thus, regulate translation in yeast during glucose starvation response, as was determined by assessing phosphorylation status of multiple mRNA-binding proteins [58]. Thirty-two RBPs in yeast strains of BY4741 background were individually C-terminally tagged with the Tandem Affinity Purification (TAP) tag and the strains were subjected to glucose deprivation. The cells were grown in YPD media at 30 °C until the optical density of 0.7–1.0 at 600 nm wavelength, and then transferred to YP (no added glucose) media for 30 min. The phosphorylation status of the tagged proteins was analyzed by a mobility shift assay comparing results for standard and Phos-tag SDS-PAGE, which slows down the mobility of phosphorylated variants. Seventeen out of thirty-two mRNA-binding proteins showed changes in phosphorylation status (electrophoretic mobility) under glucose deprivation conditions; thirteen were hyperphosphorylated and four hypophosphorylated [58]. Three major signaling pathways regulated during glucose starvation, the PKA (Protein Kinase A), Snf1p/AMPK (Sucrose Non-Fermenting / Adenosine Monophosphate-activated Protein Kinase) and TORC1, were studied for possible involvement. Bypass of Cyclic-AMP Requirement *BCY1* (encoding the catalytic subunit of PKA) or *SNF1* were deleted from TAP-tagged strains. Bcy1Δ mutants showed a lack of phosphorylation change under glucose starvation stress for Pat1p, Puf3p and Puf4p (otherwise hypophosphorylated in stress) and Pab1p and Eap1p (otherwise hyperphosphorylated in stress). In snf1Δ mutants, starvation-induced hyperphosphorylation of Bre5p, Eap1p, Nab6p, Pab1p, Pub1p, Sbp1p, Slf1p, eIF4G1/Tif4631p and eIF4G2/Tif4632p, and hypophosphorylation of Ksp1p, were abolished [58]. Phosphorylation of eIF4G1 (Tif4631p) and eIF4G2 (Tif4632p) decreased, whereas phosphorylation of Ksp1p increased under cycloheximide treatment, suggesting regulation by TORC1, which is known to be hyperactivated by the drug [237]. Immunopurifying eIF4G-TAP from WT and ksp1Δ cells and analyzing tryptic digests with liquid chromatography coupled with tandem mass spectrometry (LC-MS/MS), only nine of the thirty-three eIF4G phosphorylated residues were identified in ksp1Δ cells under stress, confirming its role in eIF4G-based regulation (Table 2; further see main text Section 6 for Ksp1p-mediated degradation of specific mRNAs during glucose starvation) [58].

IRES-reliant mechanisms were shown to be important in yeast in certain cases like ethanol/heat shock response [238] and in rapid nutritional control of translation during glucose starvation. In yeast, stress induces downregulation of translation of most cellular messages which promote morphological changes of the cells leading to invasive growth. Many invasive growth genes have unusually long 5’UTRs with the potential to form stable RNA secondary structures [239]. *YMR181c* is required for invasive growth and is the downstream ORF of a naturally occurring bicistronic cellular mRNA, the first cistron of which encodes monosaccharide catabolism-inducing zinc finger repressive transcription factor Rgm1p through RGM1/*YMR182C* ORF. Thus, it was hypothesized that invasive growth genes might be translated by an IRES-driven mechanism [66]. To test if 5’UTRs of the invasive growth genes are capable of IRES-mediated translation, these sequences were inserted after a thermodynamically stable stem-loop structure in the 5’UTR of synthetic luciferase-encoding mRNA; and the mRNAs were ApppG-capped to abolish cap-dependent initiation further, while not compromising stability. mRNAs were assessed for translation by the accumulation of luciferase activity in cell-free yeast extracts [240] in vitro and in vivo by RNA electroporation into yeast cells [241]. All seven (*YMR181c*, *GPR1*, *BOI1*, *FLO8*, *NCE102*, *MSN1* and *GIC1*) 5’UTRs promoted efficient cap-independent translation, whereas even m7GpppG-capped 5’UTR stem-loop mRNAs were poorly translated in vivo and in vitro. Cell extracts containing ~10-fold overproduced eIF4G increased invasive growth IRES activity by 10- to 20-fold. Testing 5’UTR of mRNAs from each of the two yeast eIF4G genes within the ApppG-capped hairpin reporter, the 461-nt 5’UTR of eIF4G2 showed IRES activity similar to the 5’UTRs of invasive growth genes both in vitro and in vivo. Further to determine which RNA elements are important for IRES activity, a series of deletion mutants revealed that a poly(A) tract minimally of 12 nt preceding the initiation codon of the eIF4G2 mRNA, reminiscent of the elements found in vaccinia virus cap/scanning-independent mRNAs [135], was responsible for the IRES-like behavior. 5’ poly(A)-dependent translation, in this case, was shown to be enhanced by Pab1p; intriguingly A-enriched sequence in the *Pab1* mRNA itself demonstrated Pab1p-dependent IRES activity [66] and was shown to accumulate SSU footprint pile-ups in this region [86]. Overall, starvation-induced cap-independent production of both Pab1p and eIF4G2 was shown to be able to sustain the necessary translation during prolonged periods of decreased cap-dependent initiation [66].

The so-called ‘diauxic shift’ occurs in yeast during prolonged glucose starvation, when the cells consume the earlier produced ethanol by switching to gluconeogenesis and concomitantly increasing their respiration rate, and is believed to be a consequence of a tricarboxylic acid (TCA) cycle upregulation. Adaptation and the onset of the diauxic shift were studied using FY4 yeast strain and glucose as the sole carbon source [242]. It was found that during this shift, there is a translational upregulation of the enzymes that consume acetyl Coenzyme A (acetyl-CoA) in the glyoxylate cycle and gradual translational upregulation of Nicotinamide Adenine Dinucleotide Phosphate (NADP)-utilizing enzymes, with Pentose Phosphate pathway (PP) shut down late in the transition between 4.4 and 7.4 h after the glucose depletion [242], but the exact mechanistic details of these rearrangements remain obscure.

In mammalian cells, glucose starvation induces IRES-mediated translational upregulation of the Cationic Amino acid Transporter cool-associated and tyrosine-phosphorylated protein 1 (CAT1; also known as G protein-coupled receptor kinase-associated ADP ribosylation factor GTPase-activating protein, GIT1), which is a high-affinity transporter of the essential amino acids (arginine and lysine) [243]. This was revealed when both *Cat1* mRNAs (3.4 and 7.9 knt) levels remained constant for the first 3 h of starvation and increased thereafter, with a 7-fold increase after 6 h (northern blot analysis). However, the level of CAT1 protein increased continuously and by 12 h was 30-fold higher than in non-starved cells as analyzed by western blot analysis of cell extracts probed with anti-CAT1 antibody [243]. To check if the *Cat1* mRNA IRES (known to act during amino acid starvation) was also active in glucose starvation, a vector encoding bicistronic mRNA encoding CAT1 and firefly luciferase (Luc) with 5’UTR sequence of the *Cat1* mRNA placed upstream of the second Luc-encoding cistron was constructed. C6 rat glioma cells grown in medium with or without glucose for different time intervals were transiently transfected with this vector and lysates assayed for CAT1 and Luc activity. Luc activity increased continuously until 9 h during glucose starvation and then dropped (Figure 2d). However, the increase could be stopped if the cells were supplemented with glucose at 6 h, which also resulted in Luc to CAT1 ratio return to the initial values in the subsequent 3 h [243]. Co-transfection with vectors expressing dominant-negative tagged mutant kinases GCN2 or PERK (monitored by tag western blotting) caused an increase and absence of increase of Luc translation without glucose, respectively, which pointed towards PERK-regulated response [121,243].

Glucose starvation was shown to induce regulation of p53 mRNA translation via IRES mechanisms, controlling the production of either full-length p53 or Δ40p53 [244]. Glucose starvation can increase Scaffold/Matrix Attachment Region-binding protein 1 (SMAR1) abundance in the cytoplasm (through regulation of cytoplasmic enzyme Aldo-keto reductase 1a4 (AKR1a4)) [245], which binds to and trans-activates the p53 mRNA IRES [59]. This was shown by transfecting H1299 and A549 cells with luciferase bicistronic constructs containing p53 mRNA nucleotides 1-251 in the intercistronic region. Increase in the IRES activity was observed at 4, 8, 20, and 30 h post-transfection under glucose starvation (Table 2; Figure 2b). The maximum increase in IRES activity was observed by 20 h in H1299 and 30 h in A549 cells, with a concomitant decrease in the cap-dependent translation. SMAR1 knockdown demonstrated a decrease in the levels of p53 in a small interfering (si)RNA dose-dependent manner, whereas no decrease in cap-dependent translation was observed (measured by reporter β-galactosidase activity), which served as a proxy to propose p53-specific SMAR1 action. Immunofluorescent assay indicated that SMAR1 re-localized from the nucleus to the cytoplasm at 8 h or more of starvation [59] (Table 2). Since nuclear-cytoplasmic re-localization of IRES-transacting factors is essential for cellular function, this redistribution of SMAR1 highlights its importance for p53 IRES activation [246]. Similarly, in vivo experiments in mice demonstrated a marked increase in SMAR1 levels in the thymus and liver after 24 h of caloric restriction inducing 40% decrease in the blood glucose levels, which lead to the increase of p53 and p44 (mouse Δ40p53) levels. These effects were readily reversible as observed in rescue experiments performed in vitro or in vivo in mice fed *ad libitum* after caloric restriction [246]. For p53 being a cornerstone tumor suppressor gene, deregulation of p53 consequent of nutrient starvation can contribute to cellular transformation and carcinogenesis [247,248]. As SMAR1 has been shown to be downregulated in cancers [249], a better understanding of the SMAR1 and SMAR1/p53 mediated translational control regulation might yield novel therapeutic opportunities.

Some mRNAs were shown to be translationally activated during certain pathophysiological conditions, including diabetes-associated vascular diseases or hyperglycemia. Glucose stimulation of rat insulinoma INS-1 cells for 2 h led to a significant increase in the amounts of insulin 1 and 2 and the secretory granule markers ICA512/IA-2, PC1/3 and PC2 mRNAs co-immunoprecipitated with Polypyrimidine Tract Binding Protein 1 (PTBP1). The insulin 1 and 2 and other granule cargo mRNAs, including ICA512, CgA, PC1/3 and PC2, were shown to contain PTBP1-binding sites in their 5′UTR and rely on the activity of PTBP1 as an IRES-Trans Acting Factor (ITAF) for their translation. Glucose stimulation increases the binding of PTBP1 to the 5’UTRs of mRNAs for granule components in vitro, and in vivo in case of insulin and ICA512 mRNAs [250].

## 4. Regulation by Targeting the mRNA Scanning Step

Scanning efficiency depends on the availability of the core scanning factors, such as eIF1, eIF1A, eIF2, eIF3 and eIF4A/B/G/E [78,79], as well as on the length and/or structure of 5’UTRs [251]. The activities of RNA-helicases eIF4A, Ded1p, Dbp1p [165,166,173] and RNA-binding protein eIF4B [160,252] are directly involved in making SSUs unidirectionally move 3′-ward. Ensuring this directionality can be considered as one of the largest contributors to the efficiency of the scanning process. However, it is widely appreciated that some mRNAs with very short and/or unstructured 5’UTRs may not need the ‘powered’ SSU motion to scan and reach the start codons. The net effect of factor availability on scanning efficiency is, therefore, combinatorial and not universal across all mRNAs [75,83,253].

One of the most prominent reactions to glucose removal in yeast is the loss of eIF4A from the initiating SSU complexes. While many other nutrient stress responses in yeast are channeled through the eIF2α phosphorylation pathway (also see Section 5), glucose starvation does not have a strong immediate effect on the translation of GCN4 mRNA, indicative of an initial eIF2α/eIF2B-independent response [40]. Other major possibilities, such as rapid eIF4G degradation, were excluded as well [60] (Table 3).

Yeast strains of BY (yMK36) background [254] with C-terminally TAP-tagged factors were grown in exponential phase at 30 °C with either standard YP media supplemented with 2% glucose, or synthetic complete (SC) media, and starved for 10 or 30 min and then formaldehyde-fixed. The whole cell extracts or segregated ribosomal fractions derived from sucrose density ultracentrifugation (polysome profiling) were used to TAP-isolate complexes containing individual factors [42]. Using western blotting, an increase in the levels of eIF4G and Pab1p co-purifying with eIF3 was observed in the first 10 min of starvation, but not after 30 min (Table 3; also see main text Section 6). It was concluded that glucose starvation causes a transient build-up of eIF4E, eIF4G, Pab1p, and eIF3-containing complexes, yet at later times, these complexes disassemble, leaving the CLC free to re-localize into cytoplasmic granules (Figure 4; see also Section 6 and Table 5) [42,64]. To discern if during glucose starvation the interaction between CLC and any component of the initiating SSU is (de)stabilized as an effect of translation inhibition, co-sedimentation of translation initiation factors across sucrose gradients was assayed. After 10 min of glucose starvation, eIF4G and Pab1p were found to accumulate in the SSU fraction along with eIF3, but after 30 min eIF4G and Pab1p moved away from the SSU peak into the ‘free’ protein fraction. It, thus, was concluded that the regulated step occurs while the initiating SSUs are attached to the mRNA, just before start codon recognition. Concurrently, the amount of eIF4A bound to eIF4G in the cell lysates was substantially reduced at both the 10 and 30 min of starvation as determined by western blotting, further consistent with a reduction of eIF4A co-sedimenting with ribosomal fractions [42]. Intriguingly, not much is known about what determines the selectivity of the eIF4A-mediated response, although both, selective translation during glucose stress and potentially different/selective translation (see below) depending on eIF4A function are known. It becomes paramount to understand the underlying processes as translationally-mediated repression and storage of mRNA is interconnected with the selective formation of RNA-containing cytoplasmic granules, a universal eukaryotic feature of the stress response (see more in Section 6).

Microarray analysis of polysomal material from yeast cultures starved of glucose for 10 min was performed to study global changes in translation [42]. It was observed that most of the mRNAs, which translation was maintained under glucose starvation, were involved in carbohydrate metabolism. The downregulated mRNAs included those with cell cycle, cell transport, ribosome biogenesis and transcription gene ontology, consistent with the general inhibition of growth and proliferation of yeast. For example, *HXT2* (Hexose transmembrane transporter), *HSP30* (Heat Shock Protein 30), and *MTH1* (MSN Three Homolog; negative regulator of the glucose-sensing signal transduction pathway) mRNAs exhibited increased abundance levels in polysomes following glucose starvation, whereas the *AQR1*, *CDC6*, *PCL1*, and *RPA12* mRNAs were decreased as measured by both qRT-PCR and microarray. The up- and down-regulated mRNAs were further examined, and no significant differences for polysomal abundance of mRNAs with UTR length or persistent sequence patterns in the 5’ and 3’UTRs between the groups were found. However, the (G,C) content immediately upstream of the main ORF start codons was lower in the set of up-regulated mRNAs. This observation was intriguing as it was previously reported that A-rich sequences could serve as IRES elements to promote translation initiation post glucose starvation [42,66]. The loss of eIF4A as a result of glucose starvation could favor initiation on mRNAs with non-structured 5’UTR sequences, such as what could be expected on mRNAs with lower (G,C) content. However, another tempting general model might be that other ATP-dependent RNA helicases, such as Ded1p, or the functionally related Dbp1p, would substitute for eIF4A allowing translation of these mRNAs (see Section 6 for more details). Ded1p is known to be important for the scanning of longer and more structure-prone 5’UTRs in yeast [166,173], including when eIF4A is lost from SSU complexes under glucose starvation [42], and we can expect more investigations into the dynamics of the process to connect it to the overall translational response in this stress.

Studies in mammalian (HEK 293) cells using ribosome profiling revealed global inhibition of translation caused by Rocaglamide A (RocA), which nonetheless triggered a highly selective response in certain mRNAs. RocA is known to suppress aneuploid tumor cells and targets eIF4A by increasing the affinity between eIF4A and RNA. It was shown to induce eIF4A clamping to polypurine sequences located in the 5’UTRs in an ATP-independent manner [255,256]. A similar mechanism, whereby initiating/scanning SSUs would ‘get stuck’ in certain 5’UTRs, could be imagined explaining some cases of mRNA-selective response during stress, such as those due to the loss of eIF4A.

Overall, despite much progress, the subsets of mRNAs that are specifically targeted during glucose starvation responses in yeast, and the exact mechanisms of selectivity, are not fully established. Given the critical role of eIF4A loss, and the potential function of additional translation-related helicases in these responses, an answer to this question might come from methods that can track scanning SSUs in 5’UTRs [86]. This would help to discern cases with a complete block of SSU attachment to mRNA from those where accumulation and stacking of SSUs in the 5’UTR occurs due to scanning impediments.

## 5. Regulation by Targeting Start Codon Recognition

Upon start codon recognition, eIF5 induces hydrolysis of eIF2-bound GTP to guanosine diphosphate (GDP), which triggers the release of eIF2:GDP:eIF5. The GDP then can be recycled to form eIF2:GTP again by eIF2B, a guanine nucleotide exchange factor (GEF) that displaces eIF5 and promotes TC formation [141,257]. eIF2α is an important target of regulation at the initiation phase. Phosphorylation of eIF2α by stress-responsive eIF2α kinases increases the affinity of eIF2 for eIF2B and transforms it from a substrate to a competitive inhibitor of eIF2B, thereby reducing TC levels [141,258].

There is one eIF2α kinase in yeast, General CoNtrol protein 2 (Gcn2p). Nutrient deficiency, such as removal of amino acids from the cultivation media, causes activation of Gcn2 kinase which phosphorylates eIF2α on Ser51 [16,259], leading to the depletion of active eIF2, resulting in low TC levels and deficiency of active eIF2 in cells. Decreased levels of eIF2 lead to the increase in a specific variant of uORF-mediated regulation, as exemplified by a classical mechanism controlling General CoNtrol protein 4 (*GCN4*) mRNA translation [16]. During conditions of low TC, upon termination on an upstream uORF with a special sequence, SSUs move 3’-ward as in the normal scanning, but without TC, which induces skipping of the start codons of downstream uORFs (similar to ‘leaky scanning’) until the TC is re-captured [89,157].

To study the possible implications of eIF2α phosphorylation in glucose starvation, a time-course experiment of eIF2α phosphorylation and induced *GCN4* expression was performed using the RY124 (MATα ura3-52 leu2 trp1 P180 [GCN4-lacZ, URA3]) yeast strain [60]. Cells were grown to mid-logarithmic phase in SD medium, and shifted to minimal medium containing low 0.05% *w*/*v* glucose compared to the control cells shifted to the same media, but with 2% *w*/*v* glucose. Cells were assayed after 2 h and then after every additional hour of incubation. Phosphorylation of eIF2α, as measured by immunoblotting, was found to be increased only 4 h into glucose starvation, much later than the rapid translational response occurs (see in main text Section 3, Section 4 and Section 6). Interestingly, there also was an increase in Gcn4-lacZp abundance, as measured by the fused β-galactosidase enzyme activity, between 4 to 6 h into glucose withdrawal. By 8 h of glucose starvation, eIF2α phosphorylation declined, and there was no further accumulation of Gcn4-lacZp (β-galactosidase enzyme activity). These results indicate that eIF2α phosphorylation by Gcn2p is a transient event occurring late during glucose starvation. Furthermore, the increase of *GCN4* expression and eIF2α phosphorylation by Gcn2p happened simultaneously in response to glucose starvation [60]. These data support Gcn2p function in the maintenance of glycogen levels during prolonged glucose starvation, but not in the acute response, suggesting a more global role of Gcn2p in reprogramming pathways between amino acid and glycogen metabolism (Figure 3; Table 4).

In mammalian cells, there are four known eIF2α kinases, General CoNtrol protein (GCN2), Protein Kinase R (PKR), PKR-like endoplasmic reticulum kinase (PERK) and Heme Regulated Inhibitor (HRI). Translation of the Activating Transcription Factor 4 (*ATF4*) mRNA is upregulated under cellular stress conditions following a similar mechanism as observed with yeast *GCN4* mRNA [260,261,262]. However, the eIF2α-mediated response is not as well studied for glucose starvation in mammalian cells. Ribosome profiling was performed in mammalian cells [57] to observe the fast (20, 40, and 60 min) response of neural cells (PC12 cell line derived from *Rattus norvegicus*) to Oxygen and Glucose Deprivation (OGD, a stress mimicking physiologically important conditions, such as physical load, ischemia or cancer) (Table 4) [57]. OGD imposes dramatic stress on cellular bioenergetics and mitochondrial function [57]. Strikingly and in parallel to the fast reactions in yeast, most of the changes occurred rapidly within the first 20 min and continued to ramp up during the exposure to OGD. Increased ribosome density on mRNAs upstream of annotated main coding ORFs (acORF) was observed as the most general response [57]. For many mRNAs under OGD (e.g., up to 10-fold for *Eif1a* and *Eef2k* encoding translation factors and *Junb* encoding jun B proto-oncogene), this build-up was concomitant with a decrease in acORF translation (measured by relative ribosome footprint density). Conversely, *Eif5* and *eIF1* mRNAs demonstrated increased (2-fold) synthesis from acORFs under OGD, possibly as feedback to counteract the OGD-induced start codon selection infidelity (Table 4). Overall, it is clear that, compared to yeast, in mammalian cells there is a more rapid and more complex response to glucose starvation at the level of eIF2 phosphorylation and start codon recognition, but the underlying details and the extended dynamics of the process in relation to control of other stages of initiation remain to be fully elucidated.

In metazoans, eIF2α phosphorylation plays an important role in glucose metabolism [9]. For example, control of glucose production (gluconeogenesis) in the liver is mediated by GCN2, where eIF2α phosphorylation may increase the synthesis of gluconeogenic enzymes, such as PhosphoEnolPyruvate Carboxykinase (PEPCK) or of transcription factors that activate the expression of gluconeogenic genes [9,263,264,265]. Fluctuations in glucose levels are known to control PERK activity, which leads to eIF2α phosphorylation, and increased proinsulin translation [265] (Table 4). PERK is a transmembrane eIF2α kinase that has been found in all multicellular eukaryotes as a component of ER controlled by Binding Immunoglobulin Protein (BiP). BiP is an ER chaperone protein that senses misfolded proteins, and thus, induction of its synthesis is critical for helping to restore ER homeostasis. Accumulation of unfolded proteins in the ER results in BiP dissociation from PERK and in the PERK activation, leading to the phosphorylated eIF2α and decreased TC levels [199,264]. By using tracing translation by T-cells (3T) it was shown that *BiP* mRNA harbors uORFs that are constitutively translated during stress. *BiP* 5’UTR uORFs are exclusively initiated by the non-AUG leucine codons UUG and CUG and are eIF2A-dependent [184]. An important pathological condition linked to these control mechanisms is Wolcott-Rallison syndrome, a rare human autosomal recessive genetic disorder characterized by early infancy type I diabetes, and the first human disease directly linked to translational defects. It is caused by mutations in the human PERK (*EIF2AK3*) gene [266]. Mutations in PERK prevent its catalytic activity [266], which leads to the loss of pancreatic β-cells, resulting in permanent diabetes. The disease further develops into multiple systemic disorders, including epiphyseal dysplasia, osteoporosis, and growth retardation [267]. As with PERK mutations in humans, loss of PERK function in mice by gene targeting results in a deficiency of pancreatic β-cells and, therefore, in diabetes [268,269] (Table 4).

To conclude, ‘profiling’-type methods allowed us to glean insight into the start codon recognition-mediated response during glucose starvation, both in yeast and higher eukaryotes. However, the nature of this process full dynamics, including an understanding of where and in which state the corresponding mRNAs are localized intracellularly during each stage of the response, and the interconnection of the response with control at the other initiation stages and translation phases, remain to be elucidated. Given the apparent prominence of eIF2-based responses to glucose starvation stress, an important future perspective would also be to identify the possible involvement of alternative initiator tRNA carriers, such as eIF2A, eIF2D and MCT1:DENR (multiple copies in T-cell lymphoma and density-regulated protein), as well as the possible effects on repeat-associated translation [183,184,185,186] (Figure 3).

## 6. Formation of Cytoplasmic Foci and the Fate of Inhibited mRNA

Diverting mRNA from polysomes to specialized cytoplasmic RNP granules (also called bodies or foci) allows localized control of translation and stability of the mRNA through translational masking, storage or decay decisions [53,270,271]. Despite the granule moniker, these bodies are now understood to be more droplet-like and may form by liquid-liquid phase separation [45,46,47,48]. The exact functions of RNP granules are still actively debated and researched. Underpinning their functional importance, defects in these mRNA segregation mechanisms have been implicated in many diseases, including cancer, microbial infection, diabetes and inflammatory disease [49,54,272,273,274,275,276]. Indeed, a commonly characterized response to stress is a sequestration of mRNAs encoding non-essential proteins, while the translation of mRNAs encoding stress response proteins continues [53,270].

Following glucose starvation in yeast, most mRNAs enter several distinct, but interrelated types of cytoplasmic foci: the well-studied RNA processing bodies (P-bodies) and stress granules (SGs) [43,277], or the more elusive eIF2B-bodies [51,64]. P-bodies contain untranslated mRNAs, translation repressors, the mRNA decapping machinery and 5’ to 3’ exonucleases [49,61,278,279,280,281,282,283,284]. P-bodies are conserved, having been observed in yeast, plants, nematodes, flies, and mammalian cells [53,283]. In accordance with their protein constituents, P-bodies were shown to function in mRNA decapping [61,285], nonsense-mediated decay (NMD) [286], mRNA storage [287], as well as general [44] and microRNA-mediated translational repression [288]. The composition of P-bodies can further vary considerably according to the stress type [281,289]. SGs or eIF4E, eIF4G, Pab1p bodies (EGP-bodies) contain said proteins along with some P-body components, but not decapping/degradation enzymes and are principally formed during stress. eIF2B-bodies contain the eIF2B and eIF2 complexes as the characterizing components [51,64].

In yeast, P-bodies were identified by studying the localization of mRNA decay machinery components under glucose starvation. Dcp2p is a catalytic subunit of the decapping enzyme [290], while Dhh1p activates decapping, supposedly by remodeling the 5’ end-proximal structure of the mRNA, facilitating access of the decapping enzyme to the message [61,62,291]. It has been suggested that the degradation of mRNAs might mostly happen co-translationally [292]. This might be a major processing pathway in non-stressed cells, where P-bodies are not numerous or large and Dhh1p, and the 5’ to 3’ exonuclease Xrn1p, have been found to be associated with polysomes [292,293]. For the glucose starvation stress, cells expressing GFP-tagged version of Dhh1p (yRP1724) and Dcp2p (yRP1724) were subjected to YP media without added glucose for 10 min, washed and resuspended with synthetic medium (SC) with or without glucose, and observed via confocal microscopy. P-bodies of glucose-starved cells showed an increase in number and brightness for both Dcp2p-GFP and Dhh1p-GFP compared to the non-starved cells [281] (Figure 4; Table 5).

The mechanism by which mRNA could be stored together with such high concentrations of mRNA decay components has been suggested to be based on the presence of translation initiation factors. It has been proposed that eIF4E, eIF4G, and Pab1p would form a minimal core protecting the mRNA [64]. Re-localization of these factors to P-bodies was shown in strains containing genomic GFP fusions of eIF4E, eIF4G1, eIF4G2, eIF4AI and Pab1p, grown in glucose-rich SC medium exponentially and then in SC medium without glucose for 30 min [64] (Figure 4; Table 5). Interestingly, the seemingly ubiquitous eIF3b, eIF4AI, eIF2α or eIF2Bγ did not show re-localization into P-bodies upon glucose starvation. In contrast, 10 min glucose starvation in the same system failed to induce cytoplasmic granules with eIF4E, eIF4G, and Pab1p [64]. These observations were further confirmed by sedimentation of eIF4E, eIF4G and Pab1p away from ribosomal fractions in sucrose gradients derived from the cells starved for 30 min [61,64]. Critically to the ‘protection’ hypothesis, possibilities for mRNA interactions of these factors remained seemingly intact, both with 5’ cap and 3’ poly(A), upon starvation. This was confirmed by cap and poly(A) affinity chromatography where the levels of eIF4E and copurifying eIF4G via the cap affinity column and Pab1p, eIF4G, or eIF4E via poly(A) column remained unchanged in both glucose-starved and non-starved cells [64].

SGs in yeast are often characterized by the presence of Pub1p which is an RBP involved in the post-transcriptional regulation of expression of numerous genes [294], along with the common constituents Pab1p, eIF4E, and eIF4G. Importantly, it was shown that the highly-conserved DEAD-box protein, Ded1p, is a strong candidate for modulating the composition of messenger ribonucleoprotein complexes (mRNPs) in SGs [63]. Ded1p acts as an RNA-dependent helicase or ‘RNA chaperone’ and can remodel mRNP complexes. Ded1p can function as a translational repressor when present at high concentrations; its overproduction impairs yeast cell growth and induces the assembly of P-bodies or SGs [63,295]. To examine Ded1p localization in 15 min of glucose starvation, plasmids encoding Ded1p-GFP and Pub1p-mCherry were transformed into WT yeast and the fluorescence signals tested for co-localization against the non-starved cells. Ded1p-GFP co-localized with Pub1p-mCherry confirming Ded1p is a component of yeast SGs (in similarity to its higher eukaryotes orthologs, Belle in fruit flies [296] and DDX3 in mammals [297]). Upon Ded1 overexpression, Pub1 and eIFs 4A, 4B, 4G, 4E, but not 3, 2α, 1 and 5, were found to accumulate in SGs, as measured by localizing GFP-tagged versions of these proteins [63].

Interestingly, it was shown that during glucose starvation for 15 min, increased Ded1p interaction with eIF4G facilitated the assembly of translationally repressed mRNPs that accumulated in SGs independent of the Ded1p ATPase function. The release of mRNPs from SG, though, required Ded1p ATP hydrolysis, which was confirmed by the accumulation of GFP fusions with Ded1p ATPase loss-of-function alleles (E307A and R489A) in SGs upon 15-min glucose starvation and overall more prevalent SGs with these mutants. Based on these observations the predominant hypothesis is that a Ded1p:eIF4F:mRNA complex forms and its formation stalls translation on the mRNA as the first step of SG assembly [63], whereby Ded1p acts in ‘passive’ mode and does not require ATP hydrolysis [63]. It is an intriguing insight into how SGs could be actively remodeled, possibly through either direct Ded1p helicase action, or through translation as Ded1p is known to facilitate initiation on certain mRNAs (see below in the next paragraph). Another possibility is that the ATPase/helicase activity of Ded1p prevents SG assembly on certain mRNAs, whereas on mRNAs where such activity is suppressed, Ded1p would promote SG assembly. Thus, speculatively, differential usage of Ded1p by mRNAs during active translation may create pre-requisites of their differential recruitment into SGs under stress conditions.

A combination of both ribosome profiling and reporter analyses was used to show differential in vivo usage of Ded1p and eIF4A in the translation of subsets of mRNAs in yeast [173]. It was demonstrated that most of the mRNAs were somewhat either jointly dependent on Ded1p and eIF4A, or required only one or the other of the helicases. A sizeable proportion of mRNAs which translation unusually strongly depended on Ded1p were enriched for stronger secondary structures that might impede scanning [164,173]. These secondary structures were suggested to be resolved primarily by Ded1p during the scanning of mRNAs, with auxiliary contributions from eIF4A. This was further confirmed by introducing stem-loop structures close to the 5’ cap and observing elevated Ded1p dependency [173]. Ded1p-dependent recruitment of mRNAs required eIF4E and·eIF4G in this system, at least for some mRNAs, such as *RPL41A*, *HOR7* and *SFT2-M*. Ded1p domains responsible for interactions with eIF4G or eIF4A were shown to enhance Ded1p stimulatory effects on initiation for all tested mRNAs [164], consistent with other reports where Ded1p binding to the eIF4F was shown to enhance activity in RNA unwinding assays [166]. However, Ded1p can promote initiation on mRNAs independently of eIF4E and·eIF4G, such as for *SFT2*, *SFT2-M*, *OST3*, and *CD-8.1*, indicating that it can act independently of eIF4F as well [164].

Overall, for mRNAs lacking strong local secondary structure in the 5’UTR, Ded1p might only be needed to promote eIF4F binding to the cap or initial SSU attachment, and this process requires direct interaction of Ded1p with eIF4F at the mRNA 5’ end. For mRNAs with strong structures in the 5’UTR, in addition to the above role, Ded1p might assist scanning SSUs in passing through these structures independently on its association with eIF4F. Additionally, Ded1p can interact with eIF4A, SSUs, or the mRNA [164,298]. All these possibilities for interaction and clearly different translational dependence across mRNAs for Ded1p, as well as other SG factors, create a multitude of possible scenarios of how a subset of mRNAs could be preferentially regressed into SGs depending on a momentary translational state of the cell (Figure 4; Table 5).

A third, somewhat less understood, but distinct assembly termed eIF2B-bodies exists in yeast. eIF2B-bodies are identified as round or fiber-like structures that contain subunits of eIF2B and eIF2 complexes [299]. eIF2B-bodies rapidly and reversibly form independently of SGs or P-bodies during acute glucose starvation and during the stationary growth phase when glucose is limiting [51]. The subcellular localization of all five GFP-fused eIF2B subunits (Gcn3p, Gcd7p, Gcd1p, Gcd2p and Gcd6p) and eIF2α (*SUI2*) was carefully monitored during logarithmic growth phase and upon glucose starvation for 30 min ~5-fold increase in the number of cells (up to 20–40% of all) with eIF2B-bodies containing these proteins was observed, compared to non-starved controls [51] (Table 5).

Since both SGs and eIF2B-bodies may co-exist in the longer-term glucose starvation (~30 min), it was tested if eIF2B-bodies might be a form of SGs or physically interact with SGs. Upon 30 min of glucose starvation, as assessed by fluorescence co-localization of SG protein Pub1p tagged with mCherry with GFP-tagged eIF2B or eIF2, the cells contained either eIF2B-bodies or SGs and sometimes both (with eIF2B-bodies being initially more prevalent, probably due to their faster formation). In those cells that contained both eIF2B-bodies and SGs, these assemblies could either co-localize, be next to each another, or be localized far away from each other. These results support the SG-independent nature of eIF2B-bodies [51].

The dynamics of SGs and eIF2B-bodies disassembly upon glucose repletion was also studied, furthering the conclusion that these are distinct entities. The repletion of glucose was performed by applying agarose pads with 10% glucose over cell samples on the microscope stage (resulting in the final glucose concentration around cells of ~2%). Imaging the cells with the fluorescently tagged eIF2B-body and SG components for 15 min with 15-s resolution, it was found that SGs dissolved almost immediately, by 5.1 ± 0.9 min (average ± standard deviation), while eIF2B-bodies dissolved by 7.8 ± 0.9 min post glucose repletion [51]. Interestingly, yeast strains that did not form P-bodies (edc3Δ lsm4Δc, edc3Δ pat1Δ, dhh1Δ and pat1Δ) also caused a strong inhibition of SGs, whereas strains defective in SGs (pub1Δ, pbp1Δ and eIF4GIIΔ) formed P-bodies normally [294]. Intriguingly, mutations in the human genes encoding eIF2B leading to Vanishing White Matter Disease (VWMD) have diverse effects on stress-induced assemblies with some alleles altering eIF2B bodies, and others leading to increased P-body formation [51].

The fate of mRNAs during glucose starvation can also be determined without the apparent involvement of cytoplasmic foci. One of the best-studied examples in yeast is Ksp1p-mediated phosphorylation of eIF4G, which induces the degradation of specific mRNAs [58]. To evaluate this effect, C-terminally c-Myc-tagged eIF4G (Tif4631) with an alanine substitution at residue number 24 (eIF4G-24A) undergoing phosphorylation in WT, but not ksp1Δ cells, was introduced into eIF4GΔ cells (as compared to c-Myc-tagged genome-integrated WT eIF4G). Upon 30 min of glucose starvation, eIF4G-24A demonstrated substantially lower phosphorylation level compared to WT as analyzed by mobility in standard and Phos-tag SDS-PAGE. Further, cells synthesizing eIF4G-24A exhibited significantly slower degradation of glycolytic mRNAs under glucose starvation conditions, compared to the WT as tested by qRT PCR after blocking transcription with thiolutin treatment [58]. Half-lives of *PGK1* (encodes 3-phosphoglycerate kinase) and *ACT1* (encodes actin) mRNAs, which are specifically degraded during glucose starvation, were significantly increased in the ksp1Δ cells compared to WT post thiolutin [58]. Confirming the physiological relevance of Ksp1p-dependent mRNA degradation during glucose starvation, levels of *PGK1, ACT1* mRNAs at 0, 5, 10, and 15 min of the stress without transcriptional blockade also showed cumulative decrease in WT cells, while there was no significant acceleration of decay in the ksp1Δ cells (compared to *MFA2* mRNA as stability reference). Relevance to the glucose sensing of the Ksp1p-mediated destabilization was also confirmed by demonstrating a specific increase of degradation for other glycolytic mRNAs, such as *TDH3*, *ENO2* and *CDC19* (involved in different steps of glycolysis) in WT vs. *ksp1*Δ cells [58].

Ksp1p-dependent mRNA degradation may involve known mRNA decay activator, the DEAD-box helicase Dhh1, by tethering it to glycolytic mRNAs [300,301] (Figure 4; Table 5). Using RNA-ImmunoPrecipitation (RIP) assay wherein Dhh1p-Myc was purified (via anti-c-Myc antibody) the associated RNAs were analyzed by RT-qPCR. After 30 min of glucose starvation, increased Dhh1p binding to *PGK1*, *ACT1*, *PFK1*, *TDH3*, *ENO2*, and *CDC19* mRNAs was detected, while its association with *MFA2* remained unchanged. Because Dhh1p recruitment was severely impaired in ksp1Δ cells, starvation-induced RNA-mediated Dhh1p:eIF4G interaction may outcompete that of Ded1p:eIF4G, leading to Dhh1p attachment to the eIF4G-associated mRNAs under glucose starvation conditions and their SG-independent translational repression and destabilization [58].

In similarity to yeast, P-bodies and SGs in higher eukaryotes increase in both size and number in response to glucose starvation [281]. In mammals, the phosphorylation of eIF2α is critical for the assembly of SGs, as shown by the sufficiency of a recombinant phosphomimetic eIF2α (S51D) to induce the assembly, and the inability of a recombinant non-phosphorylatable eIF2α (S51A) to promote SG assembly [302]. Belle, an ortholog of Ded1p in *Drosophila melanogaster*, is found in neuronal translationally repressed RNA granules and thereby may play a role in modulating translation [53,173]. Other P-body constituents common across yeast, insects and mammals include Ccr4p which is a part Ccr4p/Pop2p/Not1-5p mRNA deadenylase complex and performs cytoplasmic deadenylation [303].

It is proposed that SGs (EGP-bodies) found in yeast are directly analogous to mammalian SGs. eIF4E, eIF4G, and Pab1p-containing SGs in yeast contain Pub1p (nuclear and cytoplasmic polyadenylated RNA-binding protein, involved in nucleocytoplasmic mRNA transport) and Ngr1p (Negative Growth Regulatory protein, regulates cell growth), orthologues of mammalian TIA-1 and TIA-R (members of the RNA recognition motif type family of RNA-binding proteins with functions in signaling cascades regulating entry into apoptosis) found in mammalian SGs [294].

Despite much progress, the mechanisms by which certain mRNAs are preferentially included into P-bodies or SGs are still poorly understood. A major impediment in their universal identification is rapid degradation of at least a portion of the included transcripts, and thus, the commonly used oligo(dT)-based mRNA enrichment provides an inadequate and biased pattern of P-body mRNAs. To overcome this, a method called Chemical Cross-Linking coupled to Affinity Purification (cCLAP) was devised whereby in vivo formaldehyde crosslinking was used to halt enzymatic reactions (and mRNA decay) and was followed by streptavidin affinity purification of the tagged P-body constituents [283]. Particularly, Dcp2p or Sm-domain protein member Scd6p, which were predicted to regulate the stability of mRNAs encoding cell cycle progression and vesicular assembly proteins (and representing the 5’ and 3’UTR-associated complexes, respectively), were chromosomally His_6_-biotinylation sequence-His_6_ (HBH) tandem-tagged and mRNAs bound to these factors upon 10 min of glucose starvation (or CaCl_2_ or NaCl osmotic stress) enriched and sequenced. The hits identified three classes of mRNAs, stress-type independent general P-body class (likely degraded), stress-type dependent (with variable degradation rate) and stress-specific, but stable mRNAs [304]. Another key finding of this study was the identification of specific Puf5p RBP which serves a dual role in stabilization of some mRNAs (*ATP11*, ATPase assembly) and degradation of other transcripts (*BSC1,* ion transporter). Further global studies with similar approaches, therefore, can shed more light onto the dynamic inclusion of different transcripts into P-bodies and SGs [283].

In conclusion, it needs to be mentioned that though it is well understood that the fate of mRNAs in P-bodies could either be degradation or stabilization, which could be selective to mRNA type, it is still unclear whether the same ‘type’ of P-body performs both of the functions [283]. More generally, it is an intriguing question whether there is any system to the formation of the SGs and P-bodies and if they are of a ‘universal’ or ‘generic’ type or there are differently specialized sub-classes of these foci. Given the rapid nature of the underlying rearrangements and transient presence of mRNA (either due to degradation or release), approaches based on the capture of the momentary equilibrium, such as fast in vivo crosslinking or cryopreservation, or single-molecule tracking, would be needed to classify the protein and RNA composition of P-bodies, SGs and eIF2B-bodies in more detail.

## Figures and Tables

**Figure 1 ijms-20-04043-f001:**
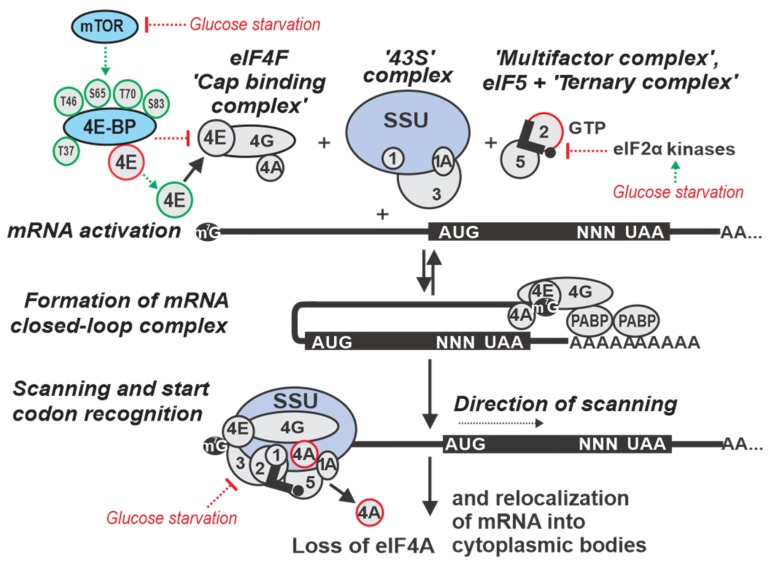
Overview of the major stages of cap-dependent translation initiation involved in the translational response to glucose deprivation. Suppressed pathways are depicted in red; activated pathways are depicted in green. SSUs shown in light state grey, initiation factors in silver, regulatory proteins and complexes not normally considered as initiation factors in light sky blue.

**Figure 2 ijms-20-04043-f002:**
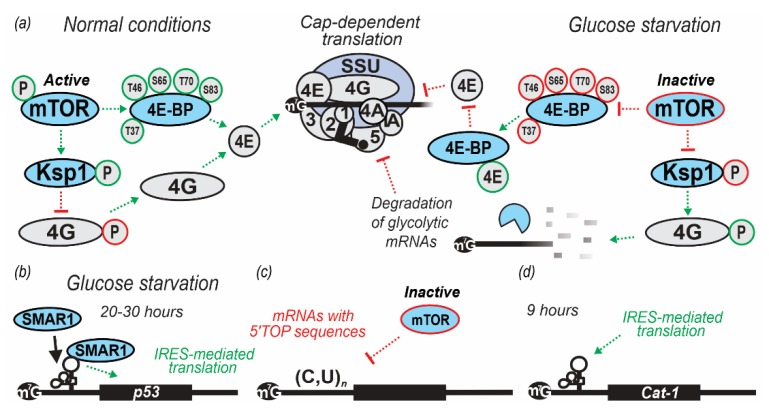
Regulation of translation during glucose starvation stress by targeting ribosomal attachment to mRNA. (a) Under optimal growth (normal) conditions, mTOR constitutively phosphorylates 4E-BP1 (T stands for threonine, S for serine), which prevents 4E-BP1 binding to eIF4E and promotes cap-dependent translation. mTOR also phosphorylates Ksp1, which promotes formation of the eIF4F complex and cap-dependent translation (green). Glucose starvation leads to inactivation of mTOR (red) which results in the accumulation of hypophosphorylated 4E-BP1, which binds eIF4E and inhibits cap-dependent translation. (b) Glucose starvation of H1299 and A549 cells for 20-30 h was shown to induce translation in mRNA-specific manner via IRES in p53 mRNA and SMAR1 ITAF. (c) Inhibition of mTOR strongly impairs (red) translation of mRNAs containing 5’TOP sequences. (d) In mammalian cells, up to 9-h glucose starvation induces IRES-mediated translational upregulation of the *CAT1* mRNA. Color coding as in Figure 1.

**Figure 3 ijms-20-04043-f003:**
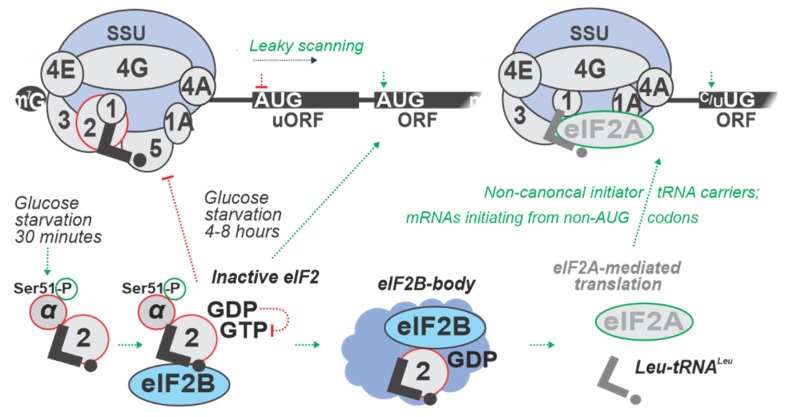
Regulation by targeting start codon recognition. Glucose starvation of ~4 h induces eIF2α phosphorylation by kinases, such as GCN2/PERK, which inhibits eIF2B recycling resulting in low levels of active eIF2 for the next round of scanning, which results in the increase of leaky scanning (between 4–8 h of starvation). Post 30 min of starvation, eIF2B-bodies begin to form, containing eIF2B and eIF2. It is also possible that during low levels of eIF2, alternate factors, such as eIF2A, are used more frequently during start codon recognition. Color coding as in Figure 1.

**Figure 4 ijms-20-04043-f004:**
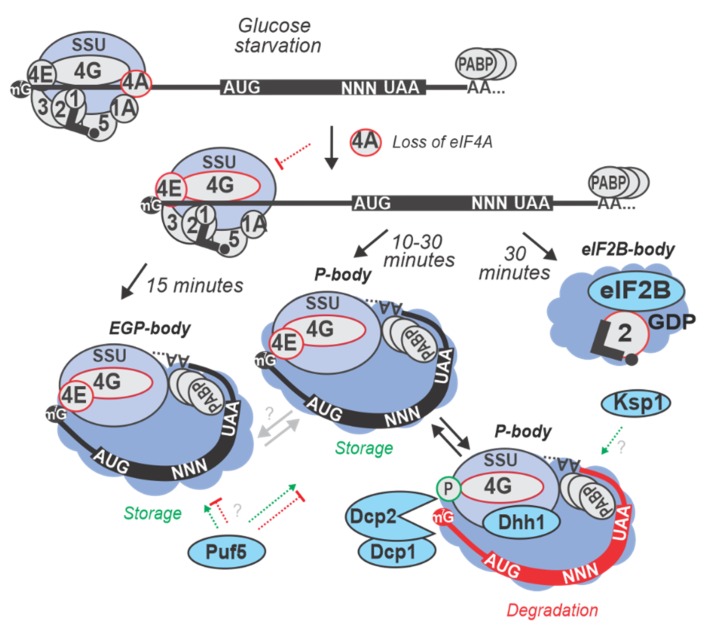
Re-localization of mRNA and translation factors into cytoplasmic foci during acute response to glucose starvation. Glucose starvation of 10 min in yeast results in the loss of eIF4A from the scanning complexes, which may stimulate utilization of other helicases, such as Ded1p or Dhh1p. EGP-bodies (SGs) are first formed at about 15 min of starvation, where the mRNA and the initiation factors are stored. Upon 30 min of glucose starvation, P-bodies (where mRNA can also be selectively degraded, such as due to phosphorylation of eIF4G by Ksp1p and recruitment of helicases like Dhh1p to glycolytic mRNAs) and eIF2B-bodies form. Regulatory RBPs, such as Puf5p, may mRNA-selectively promote storage or degradation decisions. Color coding as in Figure 1.

**Table 1 ijms-20-04043-t001:** Overview of principal studies on the regulation of translation initiation during glucose starvation.

Conditions of Stress	Organism	Main Observations and Conclusions or Models	Reference
**10 min in YP lacking glucose (0% *w*/*v* added) at 30 °C**	*S. cerevisiae* yAS2568 (W3031A)	Near-complete disassembly of polysomes observed after 10 min of starvation. Subsequent addition of glucose partially rescued polysomes within 1–2.5 min and completely restored polysomes in 10 min.	[40]
**10 min in YP with normal and low glucose (2.0, 0.7, 0.5, and 0.1% *w*/*v* added) at 30 °C**	*S. cerevisiae* yMK37 (BY4741)	Disassembly of polysomes observed when glucose levels were lower than 0.5% *w*/*v*. High concentration of glucose was determined as necessary for a yeast batch culture to maintain efficient translation.	[42]
**10, 30, 60, 120 min in YP lacking glucose (0% *w*/*v* added) at 30 °C**	*S. cerevisiae* (MATα *ura3 leu2 trp1 his3)* (1278b)	Polysomes remained low-abundant after 10 and 30 min of glucose starvation and showed partial recovery after 60 min, with further increased after 120 min.	[55]
**15 min in SC lacking glucose (0% *w*/*v* added) at 30 °C**	*S. cerevisiae* (BY4741) and EY0690 (W303 MATa *trp1-1 leu2-3 ura3-1 his3-11 can1-100*)	Ribosome profiling revealed an inversed correlation between the change in ribosome occupancy upon glucose starvation and the change in mRNA expression levels.	[56]
**20, 40, and 60 min of Oxygen (0%) Glucose Deprivation (OGD); cells maintained in DMEM (lacking glucose), at 95% N_2_ and 5% CO_2_, 37 °C.**	*Rattus norvegicus* PC12 (neural cell line)	Significant alteration of translation of approximately 3,000 genes occurred in the first 20 min of the stress (ribo-seq data).	[57]

**Table 2 ijms-20-04043-t002:** Regulation by targeting ribosomal attachment to mRNA during glucose starvation stress.

Conditions of Stress	Organism	Main Observations and Conclusions or Models	Reference
**10 min in YP lacking glucose (0% *w*/*v* added)**	*S. cerevisiae* yMK1751 (*caf20Δ*), yMK1752 (*eap1Δ*) and yMK1752 (*caf20Δ eap1Δ*) (BY4741)	Single and double mutant strains Caf20Δ and Eap1Δ exhibited polysome disassembly as a result of starvation. Therefore, Caf20Δ and Eap1Δ 4E-binding proteins concluded as not critical for inhibition of translation initiation during glucose starvation in yeast.	[42]
**30 min in YP lacking glucose (0% *w*/*v* added)**	*S. cerevisiae (*BY4741)	During glucose starvation, eIF4G was phosphorylated by the serine/threonine protein kinase Ksp1p. Thirty-three phosphorylated residues of eIF4G were identified in WT cells and nine (S17, T138, S140, T183, T399, T400, S502, S503, T712) were identified in *ksp1*Δ cells during glucose starvation. This phosphorylation is regulated by both Snf1p/AMPK and TORC1 pathways. TOR promotes anabolic, and AMPK promotes catabolic responses when cells are high or low on nutrients, respectively.	[58]
**4, 8, 20 and 30 h in DMEM lacking glucose at 37 °C**	*Homo sapiens* H1299 (p53-null lung cells derived from metastatic site lymph node) and A549 cells (lung-derived, expressing endogenous p53)	Glucose starvation was shown to induce cap-independent production of p53 and increase scaffold/matrix attachment region-binding protein 1 (SMAR1) abundance in the cytoplasm. SMAR1 was demonstrated to bind to the p53 IRES and suggested to control translation of mRNA encoding p53 isoform (Δ40p53). The increase in p53 isoform production was reversible, suggesting that transient glucose or dietary deprivation could impinge reversibly on p53 signaling, as also suggested by p53-target transactivation observed in the study.	[59]

**Table 3 ijms-20-04043-t003:** Regulation by targeting scanning of mRNA 5’UTRs during glucose starvation stress.

Conditions of stress	Organism	Main observations and conclusions or models	Reference
**10 min in SC lacking (0% *w*/*v*) glucose**	*S. cerevisiae* (BY4741)	An increase in the levels of eIF4G and Pab1p co-purifying with eIF3 was observed (transient build-up of eIF4E, eIF4G, Pab1p, and eIF3) in the first 10 min of the glucose starvation. After 30 min of starvation, this association weakened with possible re-localization of the corresponding closed-loop complex (CLC) factors and mRNAs into cytoplasmic granules. Severe reduction in the levels of eIF4G:eIF4A complex observed at both 10 and 30 min.	[42]
**20, 40, and 60 min of Oxygen (0%) Glucose Deprivation (OGD); cells maintained in DMEM (lacking glucose) at 95% N_2_ and 5% CO_2_, 37 °C.**	*Rattus norvegicus* PC12 (neural cell line)	Widespread increase of ribosome density in mRNA 5’UTRs upstream of annotated main coding ORFs, increased efficiency of termination from 20–60 min of starvation as observed by ribosome profiling.	[57]

**Table 4 ijms-20-04043-t004:** Regulation by start codon recognition during glucose starvation stress.

Conditions of Stress	Organism	Main Observations and Conclusions or Models	Reference
**8 h in SC with minimal glucose (0.05% *w*/*v* added)**	*S. cerevisiae* RY124 *(MATα ura3-52 leu2 trp1* p180 [*GCN4-lacZ*, *URA3*])	eIF2α phosphorylation by Gcn2p protein kinase is increased transiently 4 h into and stays increased up to 8 h of glucose starvation. Gcn2p function found to contribute to the maintenance of glycogen levels during prolonged glucose starvation, suggesting a link between amino acid control and glycogen metabolism.	[60]
**20, 40, and 60 min of Oxygen (0%) Glucose Deprivation (OGD); cells maintained in DMEM (lacking glucose), at 95% N_2_ and 5% CO_2_, 37 °C**	*Rattus norvegicus* PC12 (neural cell line)	uORF-mediated inhibition of annotated coding ORFs (acORF) translation at all time points. E.g., *Eif1a* and *Eef2k* mRNAs demonstrated increased uORF translation with an accompanying decrease in acORF translation. OGD induced translation from non-AUG codons, leading to an extended variability of protein isoforms, as demonstrated for *Adm*, *Bcl211*, *Fam178b*, *Ptms*, and *Ppp1r2* mRNAs. The strongest increase in translation of uORFs was observed for *Eif1a* and *Junb* mRNAs with over 10-fold increase in ribosome density in uORFs during OGD.	[57]
**24 h in DMEM lacking glucose (0% *w*/*v* added)**	*Mus musculus* (Ser51Ala *eIF2α* mutant), mouse embryonic fibroblasts (MEF) cell line	Upon low glucose, protein folding is inefficient, and PERK activated. This was demonstrated to result in an increased eIF2α phosphorylation. Lack of ternary complex (TC) was shown to inhibit translation (including of pre-pro-insulin) and lead to an increase of transcription of ER stress response genes.	[9]

**Table 5 ijms-20-04043-t005:** Re-localization of mRNAs and translation factors into cytoplasmic granules and degradation of mRNAs during glucose starvation stress.

Conditions of Stress	Organism	Main Observations and Conclusions or Models	Reference
**15 min in SC lacking (0% *w*/*v*) glucose at 30 °C**	*S. cerevisiae* BY4741 and EY0690 (W303 MATa *trp1-1 leu2-3 ura3-1 his3-11 can1-100*)	Differential re-localization of mRNAs into P-bodies and SGs was observed. mRNAs of proteins responsible for fermentation of glucose to ethanol, such as *PDC1* and involved in glycolysis and gluconeogenesis, such as *PGK1* localized in P-bodies, whereas mRNAs transcriptionally activated by Hsf1p via heat-specific elements (such as *HSP30* and *HSP26*) remained diffusely localized in the cytoplasm and highly translated. mRNA like *GLC3 (*required for glycogen biosynthesis) and *HXK1 (*Hexokinase) that are activated via stress-regulated elements independent of Hsf1p localized to both, P-bodies and SGs and were translationally repressed during glucose starvation.	[40,56]
**10 min in SC lacking (0% *w*/*v*) glucose at 30 °C**	*S. cerevisiae* Yrp1724 (*MATa leu2-3,112 trp1 ura3-52 his4-539 cup1::LEU2/PGK1pG/MFA2pG DHH1-GFP (NEO))*	P-bodies were formed and found to contain various proteins implicated in mRNA degradation. In glucose starved cells, P-bodies increased in brightness and number when stained for Dcp2p (decapping enzyme), Dhh1p (activator of decapping enzyme). An increase in the concentration of decapping activators like Lsm1p, Pat1p, Edc3p, Dcp1p, and Xrn1p exonuclease, was also observed within P-bodies upon glucose starvation stress.	[61,62]
**30 min in YP lacking glucose (0% *w*/*v* added) at 30 °C**	*S. cerevisiae (*BY4741)	Ksp1p -dependent phosphorylation of eIF4G was found to promote degradation of mRNAs encoding glycolytic proteins (*PGK1*, *TDH3*, *ENO2* and *CDC19*) by inducing Dhh1p recruitment. Ksp1p also was shown to cause rapid degradation of ribosomal protein (RP) mRNAs (e.g., *RPL37A* and *RPS8B*) during glucose starvation conditions.	[63,58]
**30 min in SC lacking (0% *w*/*v*) glucose at 30 °C**	*S. cerevisiae* (BY4741)	Accumulation of eIF4E, eIF4G1, eIF4G2 and Pab1p was observed in cytoplasmic granules after 30 min, but not 10 min of starvation.	[42,64]
**15 min in SC lacking (0% *w*/*v*) glucose at 30 °C**	*S. cerevisiae* yRP2065 (BY4741)	DEAD-box protein Ded1p (mammalian ortholog: DDX3) was found to accumulate in SGs. Ded1p was also shown to assemble translationally repressed mRNPs that accumulated in SGs independent of Ded1p ATPase activity. It was demonstrated that the Ded1p ATPase was required for the release of mRNPs from SGs.	[63]

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
