# Peer review of "Control of Translation at the Initiation Phase During Glucose Starvation in Yeast"

_ijms, 2019, doi:10.3390/ijms20164043_

Round 1
Reviewer 1 Report
This manuscript reviews a wide spectrum of knowledge on regulation of eukaryote translation, with emphasis on effects by glucose starvation. Most of thre material reviewed is from yeast, but comparisons with mammals are also done.
The overview given by this manuscript is good and clear to a nonspecialist, hence from my nonspecialist point of view it quite fulfills its purpose.
My comments below should be seen as advise for potential improvements, the authors should consider them and implement them if these imply a reasonable amount of efforts.
1. Please do a special effort to check if more material is available from other eukaryotes on the subject, and if there is, integrate it in the review.
2. In bacteria, Hecht et al 2017 Nucleic Acids Res. 2017 Apr 20;45(7):3615-3626. doi: 10.1093/nar/gkx070 estimated translation initiation from all 64 codons, and found this occurs at quantifiable levels at most codons (including at some stop codons). Is comparable information available for eukaryotes? It is likely that starvation induces noncanonical initiation, as is known for noncanonical termination at UGG codons (Chavatte et al 2003 JMB; EMBO).
3. Along that line, I suggest to broaden the comparison with prokaryotic translation, stressing differences and similarities. I would also discuss (if material is available) or suggest that future studies should include/focus on mitochondrial translation and its regulation, with questions such as: more pro- or more eukaryotic? and whether mitochondrial translation is particularly sensitive to glucose starvation.
The manuscript should be published after minor english corrections, and hopefully integrating some of the suggested materials and lines of thought.
Author Response
Response abstract
We are very grateful to the reviewers for the positive assessment of our manuscript. We accept all suggestions and made the respective corrections in Revision 1. The main additions were a more extended discussion of the prokaryotic and mitochondrial mechanisms of translational control (included in comparative manner in Section 2 – Overview of eukaryotic translation initiation control), more material on glucose starvation response in diverse eukaryotes (added to Section 3 – Regulation by targeting ribosomal attachment to mRNA) and a discussion on a possible use of non-AUG codons in glucose stress (added in Section 4 – Regulation by targeting scanning of mRNA 5'UTRs). The main changes were in the improvement of the arrangement of schematics in figures and more systematic use of indicators and colour coding of the figure elements. We have streamlined the text and introduced language edits addressing clarity and readability.
We hope that the revised version of the manuscript could be accepted for publication.
Reviewer 1
Open Review
( ) I would not like to sign my review report
(x) I would like to sign my review report
English language and style
( ) Extensive editing of English language and style required
( ) Moderate English changes required
(x) English language and style are fine/minor spell check required
( ) I don't feel qualified to judge about the English language and style
Is the work a significant contribution to the field? 3/5
Is the work well organized and comprehensively described? 4/5
Is the work scientifically sound and not misleading? 4/5
Are there appropriate and adequate references to related and previous work? 3/5
Is the English used correct and readable? 3/5
Comments and Suggestions for Authors
This manuscript reviews a wide spectrum of knowledge on regulation of eukaryote translation, with emphasis on effects by glucose starvation. Most of thre material reviewed is from yeast, but comparisons with mammals are also done.
The overview given by this manuscript is good and clear to a nonspecialist, hence from my nonspecialist point of view it quite fulfills its purpose.
My comments below should be seen as advise for potential improvements, the authors should consider them and implement them if these imply a reasonable amount of efforts.
Please do a special effort to check if more material is available from other eukaryotes on the subject, and if there is, integrate it in the review.We thank the reviewer for this suggestion and have integrated TORC1-mediated starvation response known in fruit fly larvae into the Section 3 – ‘Regulation by targeting ribosomal attachment to mRNA’.
In bacteria, Hecht et al 2017 Nucleic Acids Res. 2017 Apr 20;45(7):3615-3626. doi: 10.1093/nar/gkx070 estimated translation initiation from all 64 codons, and found this occurs at quantifiable levels at most codons (including at some stop codons). Is comparable information available for eukaryotes? It is likely that starvation induces noncanonical initiation, as is known for noncanonical termination at UGG codons (Chavatte et al 2003 JMB; EMBO).We completely agree with the reviewer that altered initiation machinery (whether due to environmental changes, infection such as viruses or genetic alterations such as cancers) can lead to more frequent use of non-AUG codons, and there are multiple experimental indications of these events available. However, no extended data is present for the glucose starvation response. We thus now include a caveat for the possible non-AUG initiation in the Section 4 – ‘Regulation by targeting scanning of mRNA 5'UTRs’.
Along that line, I suggest to broaden the comparison with prokaryotic translation, stressing differences and similarities. I would also discuss (if material is available) or suggest that future studies should include/focus on mitochondrial translation and its regulation, with questions such as: more pro- or more eukaryotic? and whether mitochondrial translation is particularly sensitive to glucose starvation.We agree with the reviewer that it is necessary to provide additional links to prokaryotic translational control and indeed, the ‘prokaryote-type’ translational mechanisms found in mitochondria can be considered as integral parts of eukaryotic cell response. We have now added major points about prokaryotic initiation control and include citations of the respective reviews, however, we would wish to avoid excessive mechanistic descriptions of these processes to keep the review well-focused on eukaryotic (cytoplasmic) translation. To address the main point about mitochondrial translation, in the Section 2 – ‘Overview of the process of eukaryotic translation initiation’ we now include a review of Couvillion, M.T., … Churchman, L.S. Synchronized mitochondrial and cytosolic translation programs Nature 2016, where a first step to dissecting mitochondrial control on a genome-wide scale was made.
The manuscript should be published after minor english corrections, and hopefully integrating some of the suggested materials and lines of thought.
We have refined the wording and corrected phrases we found inappropriate in the revised version.

Reviewer 2 Report
The review titled " Control of translation at the initiation phase during glucose starvation in yeast" by Janapala, Y et al.provides an excellent overview of the currently known mechanisms of translation initiation control in yeast during glucose starvation. The review is very thorough and is well organized into different sections describing the process of translation initiation in eukaryotes followed by different mechanisms of regulation of this step during glucose starvation.
I have only a few comments, mostly concerning Figures:
1) Fig 1: The authors should clarify what T stands for in the figure legend. Does T83 stand for threonine …. shouldn’t it be Ser 83?
2) Fig 2b: the authors mention translation to be induced denoted by green…which is not shown.
3) In line 72, the authors mention they will discuss recently developed high- throughput sequencing techniques that have been used and which can be used to solve remaining questions in the field… the authors should include a section on this.
Author Response
Response abstract
We are very grateful to the reviewers for the positive assessment of our manuscript. We accept all suggestions and made the respective corrections in Revision 1. The main additions were a more extended discussion of the prokaryotic and mitochondrial mechanisms of translational control (included in comparative manner in Section 2 – Overview of eukaryotic translation initiation control), more material on glucose starvation response in diverse eukaryotes (added to Section 3 – Regulation by targeting ribosomal attachment to mRNA) and a discussion on a possible use of non-AUG codons in glucose stress (added in Section 4 – Regulation by targeting scanning of mRNA 5'UTRs). The main changes were in the improvement of the arrangement of schematics in figures and more systematic use of indicators and colour coding of the figure elements. We have streamlined the text and introduced language edits addressing clarity and readability.
We hope that the revised version of the manuscript could be accepted for publication.
Reviewer 2
Open Review
(x) I would not like to sign my review report
( ) I would like to sign my review report
English language and style
( ) Extensive editing of English language and style required
( ) Moderate English changes required
(x) English language and style are fine/minor spell check required
( ) I don't feel qualified to judge about the English language and style
Is the work a significant contribution to the field? 5/5
Is the work well organized and comprehensively described? 5/5
Is the work scientifically sound and not misleading? 5/5
Are there appropriate and adequate references to related and previous work? 5/5
Is the English used correct and readable? 5/5
Comments and Suggestions for Authors
The review titled " Control of translation at the initiation phase during glucose starvation in yeast" by Janapala, Y et al.provides an excellent overview of the currently known mechanisms of translation initiation control in yeast during glucose starvation. The review is very thorough and is well organized into different sections describing the process of translation initiation in eukaryotes followed by different mechanisms of regulation of this step during glucose starvation.
I have only a few comments, mostly concerning Figures:
1) Fig 1: The authors should clarify what T stands for in the figure legend. Does T83 stand for threonine …. shouldn’t it be Ser 83?
We are very grateful to the reviewer for finding out this mistake and now correct it in all instances (Figures 1 & 2).
2) Fig 2b: the authors mention translation to be induced denoted by green…which is not shown.
We thank the reviewer for finding this out and now have made changes to clearly define process stages and effects with different types of arrows. Arrow with solid line and regular arrowhead define transition between stages/processes; arrows with dotted line and normal (activation) or flat (inhibition) arrowhead define regulatory influences. The arrows are colour-coded for red to depict inhibition, green to depict activation.
3) In line 72, the authors mention they will discuss recently developed high- throughput sequencing techniques that have been used and which can be used to solve remaining questions in the field… the authors should include a section on this.
We agree with the reviewer that this statement was structurally not well-represented in the manuscript. We would wish to keep the current structure of the manuscript, which is based on the functional classification of the initiation steps, and which would be interrupted by the inclusion of a dedicated section on the high-throughput methods. We thus have removed this statement from the introduction but would like to point out that appropriate references to the high-throughput methods are made throughout the entire manuscript.
